# A Flexible Eddy Current TMR Sensor for Monitoring Internal Fatigue Crack

**DOI:** 10.3390/s23239507

**Published:** 2023-11-29

**Authors:** Fei Yang, Yuting He, Xianghong Fan, Tao Chen, Teng Zhang, Binlin Ma

**Affiliations:** Aviation Engineering School, Air Force Engineering University, Xi’an 710038, China; airshinefly@163.com (F.Y.); mapleleaf1175@163.com (X.F.); starryskywide@126.com (T.C.); guangkuodedadi@126.com (T.Z.); grass6skyline@yeah.net (B.M.)

**Keywords:** FEC-TMR sensor, exciting frequency, internal crack, crack monitoring, sensitivity

## Abstract

This paper proposes a flexible eddy current TMR (FEC-TMR) sensor to monitor the internal crack of metal joint structures. First, the finite element model of the FEC-TMR sensor is established to analyze the influence of the sensor’s crack identification sensitivity with internal crack propagation at different depths and determine the optimal location and exciting frequency of the sensor. Then, the optimal longitudinal spacing and exciting frequency of the sensor are tested by experiment. The experimental results are consistent with the simulation results, which verify the correctness of the simulation model. Finally, the experiment is carried out for internal cracks of different depths to verify that the sensor can monitor internal cracks, and the crack identification sensitivity gradually decreases with the increase in the depth of the crack from the surface.

## 1. Introduction

Metal materials have excellent characteristics of high strength and are widely used in important load-bearing parts of large equipment. Affected by extreme service environments, metal structures are vulnerable to damage, which threatens the safety of the equipment. Fatigue crack is a common problem in the service of metal structures. Therefore, it is necessary to inspect the structure regularly and then find the dangerous parts of the structure and repair them in time. Among them, non-destructive technology is a commonly used method. At present, the commonly used non-destructive technologies include ultrasonic testing, magnetic flux leakage testing, penetrant testing, eddy current testing and X-ray testing.

In some applications, conventional non-destructive testing technology is inefficient and not targeted. For example, to detect the crack of the bolt hole edge, the bolt needs to be removed, and it is not possible to determine whether the bolt hole edge is cracked before disassembly. To improve the pertinence of structural inspection and maintenance, structural health monitoring technology can be used. The sensors are arranged at the dangerous parts of the structure to sense the parameters representing the structural health status, and the maintenance plan of the structure is formulated through output collection, processing and structural health status judgment. At present, sensor monitoring technologies used for structural health monitoring mainly include strain sensor monitoring technology [1,2,3], optical fiber sensor monitoring technology [4,5], comparative vacuum sensor monitoring technology [6,7], ultrasonic guided wave monitoring technology [8,9,10], smart coating sensor monitoring technology [11,12,13], eddy current sensor monitoring technology [14,15,16], acoustic emission monitoring technology [17], etc. Both strain sensor monitoring technology and optical fiber sensor monitoring technology are used to monitor the real-time load of the structure and evaluate the remaining service life of the structure. The optical fiber sensor has the advantage of anti-electromagnetic interference. Ultrasonic guided waves are mostly used for damage monitoring of composite structures, which has a wide monitoring range. They can also be used to monitor bolt looseness [18,19,20,21,22] and fatigue cracks [23,24]. Comparative vacuum sensor monitoring technology and smart coating sensor technology, which have high crack monitoring accuracy, are mainly used to monitor cracks, but they can only monitor surface cracks. Acoustic emission monitoring technology has the characteristic of high sensitivity, which can monitor the crack propagation process, but it is greatly affected by the noise environment. The eddy current sensor can be used to monitor cracks with high sensitivity, and the flexible eddy current array sensor has a large monitoring range and quantitative monitoring ability [25].

As important load-bearing parts, metal connection structures are prone to fatigue cracks at the hole edge due to the stress concentration. The cracks generated in this part have strong concealment and are difficult to find in the routine maintenance process. Therefore, many researchers have carried out a lot of work on the problem of edge crack monitoring. Among them, eddy current sensors can be used not only for non-destructive testing but also for crack monitoring, especially flexible eddy current array sensors. Sun et al. [26,27,28] also integrated the sensor with the screw of the bolt and monitored the crack at the bolt hole. Compared with the literature [29], the sensor’s induction coil is optimized. The induction coil is composed of two mutually staggered two-dimensional induction coil arrays, which can monitor the specific location of the crack at the hole edge, but the sensor does not have an anti-extrusion protective layer. Chen et al. [30] analyzed the failure form of the sensor under the bolt pre-tightening force and then integrated the sensor with the gasket to avoid the sensor crushing damage, but this increased the additional weight of the sensor.

Alexi [29] has developed a structural health monitoring (SHM) fastener which consists of an integrated eddy current sensor film. The film is combined with the metal sleeve around the fastener handle to monitor fatigue cracks at the hole. The bolt can monitor the crack propagation at the hole edge, and it is suitable for monitoring multi-layer structures. Jiao et al. [31] have integrated the eddy current array sensor in the gasket to monitor the crack at the bolt hole edge, which can effectively monitor the crack initiation and propagation. At that time, the integration method changed the stress form of the bolt structure. Liu et al. [32] have developed a new type of built-in distributed EC sensor network for composite connection structures. The sensor is arranged in the gap between the bolt hole and the bolt rod to monitor the damage at the edge of the composite bolt hole, which can monitor the generation of cracks at the edge of the composite structure hole. As the cracks propagate radially along the hole edge, the signal of the sensor’s induction coil changes significantly. He et al. [33] have proposed an annular eddy current array sensor with a reference channel. The signal of the sensor monitoring channel is corrected through the reference channel. Under the temperature coupling effect, the sensor can effectively monitor the crack propagation at the hole edge, but the sensor does not consider the role of bolts. Song et al. [34] have used flexible eddy current array sensors to study the performance of crack monitoring under corrosion, oil immersion, UV radiation and vibration environments. None of the sensors failed during the test, and the impact of bolts was not considered in monitoring hole edge cracks. For ferromagnetic materials, the literature [35] has proposed an annular eddy current array sensor with a compensation coil. The effect of temperature is eliminated through the compensation coil. The effect of cracks on the output characteristics of the sensor is far greater than that of the load on the output characteristics of the sensor. Therefore, the sensor can realize quantitative monitoring of cracks under the effect of load and temperature variation.

Although great progress has been made in the crack monitoring technology of metal connection structures, there are still many problems to be solved. For multi-layer metal connection structures, fatigue cracks not only occur on the surface of the structure, but also may occur inside the structure, and the internal fatigue cracks are hidden and difficult to find in the conventional visual inspection. At the same time, the bolt cannot be removed when the bolt hole edge crack is monitored. If the bolt is removed, on the one hand, it will introduce a large amount of work; on the other hand, it will also introduce new damage in the disassembly and installation process. The benefits of structural health monitoring will not be achieved. To obtain a higher signal-to-noise ratio, a conventional flexible eddy current array sensor has a high exciting frequency. Affected by skin effects, the sensor can only monitor surface cracks. To improve the crack monitoring depth, the skin depth of the eddy current is increased by reducing the exciting frequency, and then the disturbed magnetic field is measured by using a magneto-resistive sensor to monitor the internal crack propagation. Considering the squeezing effect of the bolt, the sensor is arranged around the bolt hole to prevent the sensor from being crushed and damaged during installation. In this paper, an FEC-TMR sensor is proposed. Firstly, the electromagnetic finite element model of the FEC-TMR sensor and the tested structure is established to determine the sensor’s optimal location and exciting frequency. Then, the crack identification sensitivity of the FEC-TMR sensor is tested for internal cracks of different depths at different locations and different exciting frequencies by prefabricating a certain length of the crack, and the optimal exciting frequency and optimal longitudinal spacing are determined to verify the correctness of the simulation model. Finally, fatigue crack monitoring tests are carried out for internal cracks of metal connection structures with different depths.

## 2. FEC-TMR Sensor

### 2.1. Sensor Structure

The FEC-TMR sensor consists of rectangular exciting coils and two TMR sensors. The TMR sensors are arranged above the exciting coils, as shown in Figure 1.

The length of the short side of the inner coil of the exciting coils is 10 mm, and the length of the long side is 30 mm. There are 11 exciting coils in total. The spacing between adjacent exciting coils is 0.3 mm, and the width of each exciting coil is 0.2 mm. It can be seen that the TMR sensor is located above the exciting coils, and the longitudinal distance is d.

### 2.2. Internal Crack Monitoring Principle

First, according to Equation (1), it can be seen that the skin depth of the eddy current will decrease with the increase in exciting frequency.
(1)$$\delta =\sqrt{\frac{1}{\pi f\mu \sigma }},$$
where δ is the eddy current skin depth. f is the exciting frequency. μ is the magnetic permeability of the metal conductor. σ is the conductivity of the metal conductor.

Taking AL2024-T4 aluminum alloy as an example, the conductivity is 1.74 × 10^7^ S/m, and the relative permeability is 1. The variation trend of the skin depth of the eddy current with the exciting frequency is shown in Figure 2.

When there is no crack in the metal conductor, the distribution of the eddy current is concentrated in the area below the exciting coils, and the propagation of the crack will disturb the distribution of the eddy current. According to the variation trend of eddy current skin depth in Figure 2, when the eddy current sensor works under low-frequency conditions, the skin depth of the eddy current increases; thus, internal cracks can be monitored.

When the crack propagates under the exciting coils in the process of propagation, the eddy current under the exciting coils will flow along the crack surface, as shown in Figure 3. When there is no crack, the eddy current under the exciting coil is assumed to flow along the positive *y*-axis direction, and a disturbed eddy current will be formed by cracks during crack propagation. To simplify the analysis, the disturbed eddy current is regarded as a line current flowing along the *x*-axis direction.

According to the distribution characteristics of the eddy current and the flow direction of the exciting current, it can be judged that the magnetic field formed by the exciting current and the secondary magnetic field formed by the eddy current will form a large magnetic field in the *x*-axis and *z*-axis directions. If the magnetic field magneto-resistive sensor monitors the *x*-axis and *z*-axis directions, it will measure a large background magnetic field. The disturbed eddy current flows along the *x*-axis direction; thus, the disturbed magnetic field will be formed on the *y*-axis and *z*-axis. Therefore, it is the best choice to measure the *y*-axis magnetic field caused by the disturbed eddy current.

### 2.3. Crack Identification Sensitivity

The magneto-resistive sensor selected in this paper is the TMR2901. The sensitivity of this sensor varies widely and ranges from 20 mV/V/Oe to 27 mV/V/Oe. Therefore, it is difficult to ensure the consistency of the sensor in the application process. To ensure the consistency of the sensor, the sensitivity of each TMR sensor can be tested, but the test cost is high. To ensure sensor consistency and reduce the test cost, this paper proposes a method to characterize sensor sensitivity by using a reference quantity. This method can suppress the inconsistency of the TMR sensor sensitivity and the external circuit of the TMR sensor.

In this paper, reference exciting coils are designed. The position of the sensor and the reference exciting coils remains fixed, and the output of the TMR sensor is connected through the operational amplifier module. When the sensor is used to monitor cracks, if the external magnetic field is $B_{0}$, and the sensitivity of the TMR sensor is $k_{0}$, then the output voltage of the TMR sensor $V_{0}$ is equal to $k_{0}B_{0}$. If the amplification factor of the operational amplifier module is $k_{1}$, then the output signal amplified by the operational amplifier is $k_{1}k_{0}B_{0}$. When the sensor is placed at a fixed position above the reference exciting coils, the magnetic field intensity generated by the exciting coils is $B_{1}$, the output signal of the TMR sensor is $V_{1}=k_{0}B_{1}$ and the output signal through the operational amplifier is $k_{1}k_{0}B_{1}$. The output voltage of the TMR sensor amplified by the operational amplifier under the magnetic field of the reference exciting coils is taken as the reference signal. When the sensor is used to monitor cracks, the amplitude of the monitoring signal is divided from the amplitude of the reference signal to obtain a reference coefficient k.
(2)$$k=\frac{V_{0}}{V_{1}}=\frac{k_{1}k_{0}B_{0}}{k_{1}k_{0}B_{1}}=\frac{B_{0}}{B_{1}},$$

It can be seen that the introduction of a reference coefficient can eliminate the influence of the sensitivity difference of the TMR sensor. The relative trans-impedance of the TMR sensor is defined as the ratio of the output signal of the TMR sensor after the operational amplifier to the sampling voltage output signal of the exciting current.
(3)$${\dot{Z}}_{Rr}=\frac{{\dot{V}}_{ir}}{{\dot{V}}_{Ir}}=\frac{{\dot{k}}_{1}\cdot {\dot{k}}_{0}\cdot {\dot{B}}_{0}}{\dot{I}\cdot r\cdot {\dot{k}}_{2}}=A_{Rr}\cdot e^{j(\theta_{1}-\theta_{2})},$$
where
(4)$$A_{Rr}\propto \frac{k_{1}\cdot k_{0}}{r\cdot k_{2}},$$
where ${\dot{Z}}_{Rr}$ is the relative impedance. $A_{Rr}$ is the relative trans-impedance amplitude, and $\theta_{1}-\theta_{2}$ is the phase difference between two signals.

When the TMR sensor is located above the reference exciting coils, the relative trans-impedance of the TMR sensor is:(5)$${\dot{Z}}_{Rrr}=\frac{{\dot{V}}_{irr}}{{\dot{V}}_{Irr}}=\frac{{\dot{k}}_{1}\cdot {\dot{k}}_{0}\cdot {\dot{B}}_{1}}{\dot{I}\cdot r\cdot {\dot{k}}_{2}}=A_{Rrr}\cdot e^{j(\theta_{1r}-\theta_{2r})},$$
where
(6)$$A_{Rrr}\propto \frac{k_{1}\cdot k_{0}}{r\cdot k_{2}},$$

So, the ratio of two relative trans-impedances is:(7)$$\frac{{\dot{Z}}_{Rr}}{{\dot{Z}}_{Rrr}}=\frac{A_{Rr}}{A_{Rrr}}\cdot e^{j(\theta_{1}-\theta_{2}-\theta_{1r}+\theta_{2r})},$$

The variation of trans-impedance is used to characterize the crack propagation. Therefore, the crack identification sensitivity can be expressed as:(8)$$\begin{array}{ll} S_{Cr} & =|\frac{{\dot{Z}}_{Rr}}{{\dot{Z}}_{Rrr}}-\frac{{\dot{Z}}_{Rr_{0}}}{{\dot{Z}}_{Rrr}}|=\left|\frac{A_{Rr}}{A_{Rrr}}\cdot e^{j(\theta_{1}-\theta_{2}-\theta_{1r}+\theta_{2r})}-\frac{A_{Rr_{0}}}{A_{Rrr}}\cdot e^{j(\theta_{10}-\theta_{20}-\theta_{1r}+\theta_{2r})}\right|\times 100\%\\ & =\frac{1}{A_{Rrr}}\cdot \sqrt{A_{Rr}{}^{2}+A_{Rr_{0}}{}^{2}-2\cdot A_{Rr}\cdot A_{Rr_{0}}\cdot \cos[(\theta_{1}-\theta_{2})-(\theta_{10}-\theta_{20})]}\times 100\% \end{array},$$

It can be seen that the crack identification sensitivity can be calculated by measuring the ratio and phase difference between the output signal of the TMR sensor after the operation amplifier and the sampled voltage output signal of the exciting current and the relative impedance of the TMR sensor when the TMR sensor is above the reference exciting coils. Therefore, when using the same TMR sensor and the operational amplifier circuit board to test the sensitivity of sensor crack identification, it is not necessary to measure the relevant parameters of the circuit and the sensitivity of the TMR sensor. Therefore, the method can effectively ensure the consistency of measurement results.

### 2.4. Reference Excitation Coil

In order to ensure the consistency of measurement results, the reference exciting coils are proposed in this paper. The reference exciting coils are square in shape and consist of 11 exciting coils. The width of the exciting coil is 0.2 mm, and the spacing between adjacent exciting coils is 0.3 mm. The side length of the innermost square exciting coil is 102 mm, and that of the outermost circle is 108 mm. The reference exciting coils are printed on the FR4 board with a thickness of 0.3 mm, and the distance between the square exciting coils and the PCB edge is 1 mm. The size of the PCB board of the reference exciting coils designed in this paper is shown in Figure 4.

To fix the TMR sensor in the upper area of the left middle position on the reference exciting coils, and the sensitive axis of the TMR sensor is along the flexible *x*-axis direction of the reference exciting coils to test the magnetic field intensity in the *x*-axis direction generated by the reference exciting coils, a fixture needs to be made to fix the reference exciting coils in a groove of the fixture, and a groove for placing the TMR sensor is set on the other side of the fixture. The fixture is shown in Figure 5, and the experimental site is shown in Figure 6.

To analyze the variation trend of the sensor crack identification sensitivity with the crack propagation under different exciting frequencies, the multi-frequency exciting method is adopted. The sine signals of different frequencies are synthesized, and then the synthetic waveform is amplified by the power amplifier module and applied to the exciting coils. The adopted synthetic exciting signal is a synthetic waveform composed of four frequency sinusoidal waveforms whose frequencies are, respectively, 0.5 kHz, 1 kHz, 2 kHz and 3 kHz, as shown in Figure 7.

It can be seen that the period of the synthesized waveform is 2 ms. Therefore, the frequency of the exciting signal is 0.5 kHz. In this paper, the principle of phase-locked amplification is used to extract the amplitude and phase of each frequency waveform and calculate the crack identification sensitivity.

## 3. Internal Fatigue Crack Monitoring Simulation

### 3.1. Finite Element Model

In this section, simulation analysis is carried out for the two-layer bolted structure to analyze the eddy current distribution law under different exciting frequencies. The simulation model of the FEC-TMR sensor and the tested structure is shown in Figure 8, and the finite element simulation model was established using COMSOL 5.4 multi-physics finite element software.

Figure 8 shows that the rectangular exciting coil is arranged near the bolt, and there is no contact with the bolt. The tested material selected in this section is AL2024-T4 aluminum alloy. The thickness of the top tested piece H_1_ is 2 mm, 3 mm and 4 mm, respectively, and the thickness of the bottom tested piece H_2_ is 2 mm. The simulation parameters are shown in Table 1, and the simulation model is shown in Figure 9.

### 3.2. The Influence of Crack Propagation on the Crack Identification Sensitivity

The crack identification sensitivity of the test point at different longitudinal distances d and heights z of the test point is studied. Considering that there is a certain error in the position of the sensor in the actual pasting process, the longitudinal distance is selected as 1 mm to 6 mm, and the interval is 1 mm. Meanwhile, the sensor is placed directly above the exciting coils. These two sensors are symmetrically distributed, and, in order to avoid duplication of work when conducting the experimental study, only one side of the TMR sensor is used to carry out the study.

The exciting frequency of the exciting coils is 0.5 kHz, 1 kHz, 2 kHz and 3 kHz. Through crack propagation, the crack identification sensitivity at the test point is obtained. The height of the test point is 0.2 mm. The simulation results are shown in Figure 10.

Figure 10 shows that the crack identification sensitivity increases first and then decreases with the crack propagation. When the longitudinal distance is 1 mm and 2 mm, the crack identification sensitivity reaches the maximum when the crack propagates to 7 mm. When the longitudinal distance is 3 mm to 6 mm and the crack propagates to 7 mm, the crack identification sensitivity reaches the maximum.

According to Figure 10, with the increase in the longitudinal distance under the same exciting frequency, the crack identification sensitivity first increases and then decreases. When the longitudinal distance is fixed, the sensitivity of the sensor to crack identification increases first and then decreases with the increase in the exciting frequency.

The exciting frequency corresponding to the maximum sensitivity is obtained under the different longitudinal distances, as shown in Table 2.

Similarly, the exciting frequency corresponding to the maximum sensitivity under the longitudinal distance when the height of the test point is 0.3 mm and 0.4 mm is shown in Table 3 and Table 4.

From the influence of the exciting frequency and the longitudinal distance under different heights of the test points on the crack identification sensitivity, it can be seen that when the exciting frequency and the longitudinal distance are fixed, the crack identification sensitivity of the test point gradually decreases with the increase in the height of the test point. When the height of the test point is fixed, the maximum crack identification sensitivity first increases and then decreases with the increase in the longitudinal distance. At different test point locations, the maximum value is obtained when the longitudinal distance is 3 mm and the exciting frequency is 2 kHz.

To further analyze the disturbance effect of crack propagation on eddy current distribution, the exciting frequency is set as 1 kHz in the post-processing of the 3D simulation model, and then the eddy current distribution diagram is analyzed under different crack lengths, as shown in Figure 11.

According to Figure 11, there is a certain disturbed eddy current at the crack tip when the crack length is 0.5 mm, but it has little effect on the eddy current distribution in the area below the exciting coils. When the crack length is 2.5 mm, it is obvious that the disturbed eddy current at the crack tip is large. The eddy current in the area below the crack flows to the left along the crack surface and around the crack tip to the upper area of the crack. As the crack propagates from the left edge of the exciting coils to the right edge, the disturbed eddy current at the crack tip shows an increasing trend. When the crack propagates to 7.0 mm, the eddy current at the crack tip also shows an increasing area. At this time, it can be seen that part of the eddy current in the area below the crack flows along the crack and around the crack tip to the area above the crack, and part of the eddy current flows along the crack to the left bolt hole edge, and finally flows into the eddy current under the left excitation coil. As the crack continues to propagate, the flow of eddy currents in these two parts becomes more obvious, and the maximum eddy current density at the crack tip shows a decreasing trend. As the crack propagates, the disturbed eddy current will gradually increase along the crack. As the crack propagates to a certain length, part of the disturbed eddy current will continue to propagate to the right along the crack surface, and part of the disturbed eddy current will propagate to the left along the surface and flow to the area below the left exciting coils along the hole edge. The crack identification sensitivity at the test point will first increase and then decrease.

Similarly, when the thickness of the top tested piece is 3 mm and 4 mm, respectively, the longitudinal distance and exciting frequency corresponding to the maximum crack identification sensitivity under different heights of test points are simulated, as shown in Table 5, Table 6, Table 7, Table 8, Table 9 and Table 10.

### 3.3. Sensor Position and Exciting Frequency Optimization

According to the simulation results, when the longitudinal distance is 1 mm and 2 mm and the crack propagates to 6.5 mm, the crack identification sensitivity reaches the maximum value. When the longitudinal distance is 3 mm to 6 mm and the crack propagates to 7 mm, the crack identification sensitivity reaches the maximum.

It can be seen that when the exciting frequency is 2 kHz and the longitudinal distance is 3 mm, the crack identification sensitivity reaches the maximum value; thus, it can be judged that the optimal exciting frequency of the test point is between 1 kHz and 3 kHz. In the simulation model, the exciting frequency is taken as the scanning parameter, and the scanning step is 0.1 kHz. The range is 1 kHz to 3 kHz. When the crack propagates to 7 mm, the crack identification sensitivity under different heights is obtained, as shown in Figure 12.

When the test point is at different heights and the exciting frequency is 1.9 kHz, the sensitivity of the crack identification rate of the sensor reaches the maximum value.

To verify the simulation results, a crack whose length is 7 mm is prefabricated at the hole edge, and then the sensor is installed on the tested structure to control the longitudinal distance, and then it measures the crack identification sensitivity under different exciting frequencies. The experiment site is shown in Figure 13, and the experiment results are shown in Figure 14.

According to the experiment results, when the longitudinal distance is 3 mm and the exciting frequency is 2 kHz, the crack identification sensitivity reaches the maximum value. Through the optimization of the exciting frequency of the test point, the maximum value is obtained at the exciting frequency of 1.9 kHz, which is consistent with the simulation results, further verifying the correctness of the simulation.

According to the results in Table 5, Table 6, Table 7, Table 8, Table 9 and Table 10, when the thickness of the top tested piece is 3 mm and 4 mm, the crack identification sensitivity at the test point reaches the maximum value when the exciting frequency is 1 kHz; thus, it can be judged that the optimal exciting frequency is between 0.5 kHz and 2 kHz. Taking the exciting frequency as the scanning parameter, the scanning interval is 0.5 kHz to 2 kHz, and the scanning step is 0.1 kHz. The crack identification sensitivity at the test point under different exciting frequencies when the crack propagates to 7 mm is obtained, as shown in Figure 15.

It can be seen that when the metal thickness of the top tested piece is 3 mm, the exciting frequency of the maximum crack identification sensitivity at the test point is 1.4 kHz. When the thickness of the top tested piece is 4 mm, the exciting frequency of the maximum crack identification sensitivity at the test point is 1.2 kHz.

Similarly, to verify the correctness of the simulation analysis results, experimental verification is carried out to verify the crack identification sensitivity at different longitudinal distances and different exciting frequencies and the corresponding exciting frequency of the maximum crack identification sensitivity. The experiment results are shown in Figure 16 and Figure 17.

It can be seen from Figure 16 that when the thickness of the top tested piece is 3 mm and the longitudinal distance is 1 mm and 2 mm, the crack identification sensitivity will reach the maximum value when the exciting frequency is 2 kHz. The crack identification sensitivity will reach the maximum value when the longitudinal distance is 3 mm to 6 mm and the exciting frequency is 1 kHz. When the thickness of the top tested piece is 4 mm, the crack identification sensitivity reaches the maximum value when the exciting frequency is 1 kHz under different longitudinal distances. Through the optimization of frequency, it is found that the maximum value is obtained at the exciting frequency of 1.4 kHz when the thickness of the top tested piece is 3 mm. When the thickness of the top tested piece is 4 mm, the maximum value of the crack identification sensitivity is obtained at the exciting frequency of 1.2 kHz. The experiment results are consistent with the simulation results, which further verifies the correctness of the simulation.

To ensure greater crack identification sensitivity, it is necessary to make the disturbed eddy current caused by the crack larger. It is found that the optimal exciting frequency decreases gradually with the increase in the thickness of the top tested piece. By reducing the exciting frequency and increasing the skin depth of the eddy current, the penetration ability of the eddy current can be improved, and then the crack monitoring sensitivity of the sensor can be improved.

## 4. Internal Fatigue Crack Monitoring Experiment

### 4.1. Experiment System

To verify that the FEC-TMR sensor has the ability to monitor internal fatigue cracks, this section carries out online fatigue crack monitoring experiments. The experiment system consists of a signal source, a power amplifier module, an FEC-TMR sensor, a two-layer metal connection structure (Figure 18), a signal conditioning module, a signal acquisition and processing system and an MTS810 material testing machine. The experiment site is shown in Figure 19.

To measure the length of the crack, a ruler is pasted on the hole edge of the front of the structure. The accuracy of the ruler is 0.1 mm. Meanwhile, to control the crack propagation along one side, an initial pre-crack is prefabricated at the right side of the lower bolt hole on the front of the structure. The length of the defect is 1 mm, and the width is 0.2 mm. Therefore, the variation trend of the crack identification sensitivity of TMR1 is observed, and the crack length during the crack propagation process is recorded. Especially, when the crack identification sensitivity of the sensor reaches the maximum value, the crack length is recorded.

In this experiment, the crack monitoring characteristics of the sensors under the top tested piece thicknesses of 2 mm, 3 mm and 4 mm are studied. The placement of the sensors is determined in Section 3.3. Meanwhile, to study the crack monitoring characteristics of the sensors under different exciting frequencies, multi-frequency exciting and single-frequency exciting (the frequency is determined in Section 3.3) are used for different thicknesses of the top tested piece.

### 4.2. Experiment Results

#### 4.2.1. The Thickness of the Top Tested Piece Is 2 mm

When the thickness of the top tested piece is 2 mm and the exciting coils are excited by multiple frequencies or a single frequency, the variation trend of the sensor’s sensitivity to crack identification with the crack propagation is as shown in Figure 20.

It can be seen from the crack monitoring results in Figure 20a that, with the crack propagation, the sensitivity of the TMR sensor to crack identification under different exciting frequencies increases first and then decreases. When the crack propagates to 7.1 mm, the sensitivity of the TMR sensor to crack identification reaches the maximum. Meanwhile, it is found that the maximum crack identification sensitivity of the TMR sensor increases first and then decreases with the increase in frequency, and the maximum value is 7.35% when the exciting frequency is 2 kHz. When single-frequency exciting is used and the crack propagates to 7.1 mm, the crack identification sensitivity of the sensor reaches the maximum value, which is 7.55%.

#### 4.2.2. The Thickness of the Top Tested Piece Is 3 mm

When the thickness of the top tested piece is 3 mm and the exciting coils are excited by multiple frequencies or a single frequency, the variation trend of the sensor’s sensitivity to crack identification with the crack propagation is as shown in Figure 21.

According to the crack monitoring results in Figure 21a, the sensitivity of the sensor to crack identification gradually increases with the increase in the exciting frequency as the crack propagates. When the crack propagates to 7.3 mm, the crack identification sensitivity reaches the maximum value. With the crack’s continuous propagation, the sensor’s sensitivity to crack identification presents a downward trend. With the increase in the exciting frequency, the maximum sensitivity of the sensor to crack identification shows a trend of increasing first and then decreasing. When the exciting frequency is 1 kHz and 2 kHz, the maximum sensitivity of the sensor to crack identification is similar, about 2.72%. When the exciting frequency is 1.4 kHz, the sensor also obtains the maximum value of 2.75% when the crack length is 7.3 mm. With the crack continuing to propagate, the crack identification sensitivity of the sensor shows a downward trend.

#### 4.2.3. The Thickness of the Top Tested Piece Is 4 mm

When the thickness of the top tested piece is 4 mm and the exciting coil is excited by multiple frequencies or a single frequency, the variation trend of the sensor’s crack identification sensitivity with crack propagation is as shown in Figure 22.

According to the crack monitoring results in Figure 22a, the crack identification sensitivity of TMR increases first and then decreases with crack propagation. When the crack length is 7.3 mm and when the exciting frequency is 1 kHz, the crack identification sensitivity is the maximum, which is 1.13%. When single-frequency exciting is used, when the crack propagates to 7.3 mm, the sensitivity of the sensor to crack identification reaches the maximum value, which is 1.19%. With crack propagation, the sensitivity of crack identification shows a decreasing trend.

### 4.3. Analysis of Experimental Results

Through the internal fatigue crack monitoring experiments under different thicknesses of the top tested piece, it is found that the FEC-TMR sensor can monitor the propagation of internal fatigue cracks. With the increase in the thickness of the top tested piece, the sensitivity of the sensor to crack identification shows a decreasing trend. When using single-frequency exciting, the signal-to-noise ratio of the variation trend of crack identification sensitivity is significantly greater than that during multi-frequency exciting.

According to the variation trend of the sensitivity of the two different sensors to crack identification under the top tested pieces with different thicknesses, the results are as shown in Table 11.

With the increase in the thickness of the top tested piece, the sensitivity of the sensor to crack identification under the same exciting frequency shows a downward trend, mainly due to the skin effect. The eddy current in the metal conductor shows an exponential downward trend along the thickness direction. Therefore, the thicker the top tested piece is, the smaller the eddy current on the bottom tested piece is. Therefore, the smaller the disturbed magnetic field formed by the disturbed eddy current is, the smaller the sensitivity of the sensor to crack identification is. Meanwhile, it can be seen that when the crack propagates to about 7 mm, the crack identification sensitivity of the TMR sensor reaches the maximum value, which is consistent with the simulation results (Table 2, Table 3, Table 4, Table 5, Table 6, Table 7, Table 8, Table 9 and Table 10), and the preliminary quantification of the crack length can be judged according to the maximum sensitivity of the TMR sensor.

## 5. Conclusions

Considering that the internal crack of the metal connection structure is a hidden hazard source, this paper proposes an FEC-TMR sensor to monitor the internal fatigue crack at the hole edge of the metal connection structure, and the following conclusions are obtained:(1)The reference exciting coils are a good choice for suppressing the inconsistency of the TMR sensor sensitivity and the external circuit;(2)The FEC-TMR sensor can monitor the internal cracks of bolt-connected structures, and the crack length can be preliminarily quantified according to the crack identification sensitivity of the TMR sensor when it reaches the maximum value;(3)With the increase in the thickness of the top tested piece, the sensitivity of the sensor to crack identification shows a downward trend, and the corresponding exciting frequency decreases gradually when the sensitivity reaches the maximum;(4)Compared with multi-frequency exciting, single-frequency exciting has a higher signal-to-noise ratio. When monitoring deep internal cracks, it is necessary to consider using single-frequency exciting to obtain a higher signal-to-noise ratio.

The results of this paper can be used to monitor the bolted connection structures that are difficult to detect and ensure the safe operation of the structures in engineering applications.

## Figures and Tables

**Figure 1 sensors-23-09507-f001:**
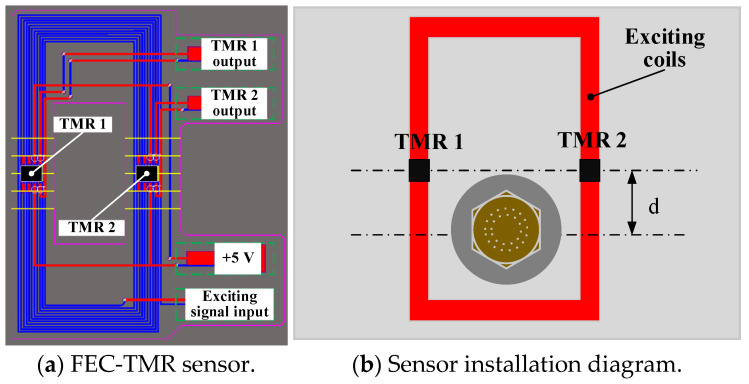
FEC-TMR sensor and its installation diagram.

**Figure 2 sensors-23-09507-f002:**
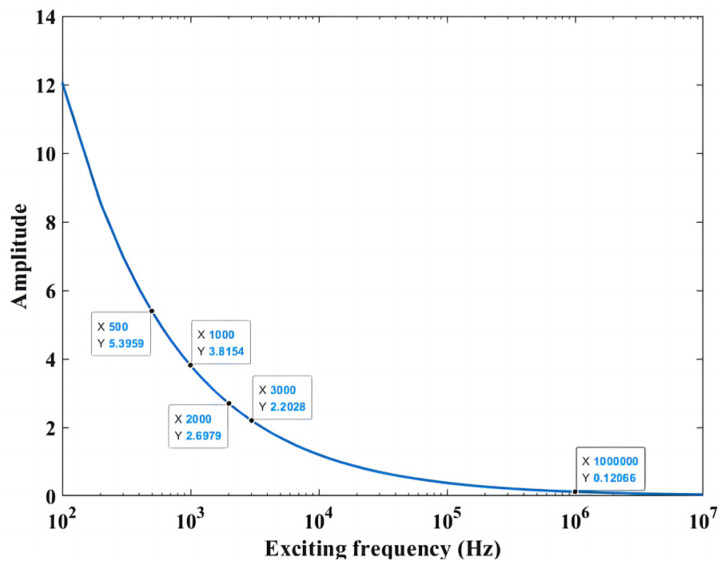
Variation trend of eddy current skin depth with exciting frequency.

**Figure 3 sensors-23-09507-f003:**
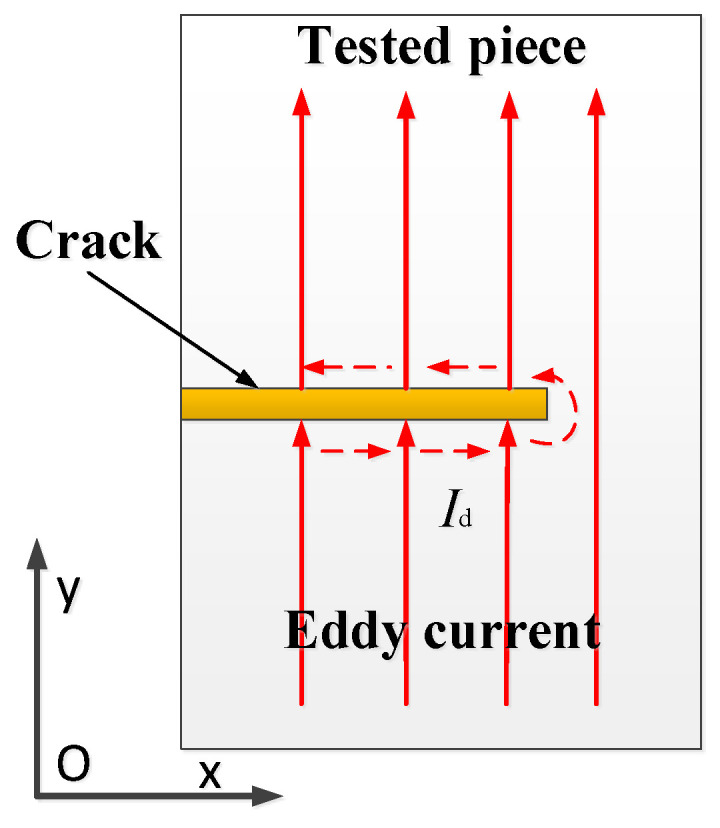
Disturbance of eddy current.

**Figure 4 sensors-23-09507-f004:**
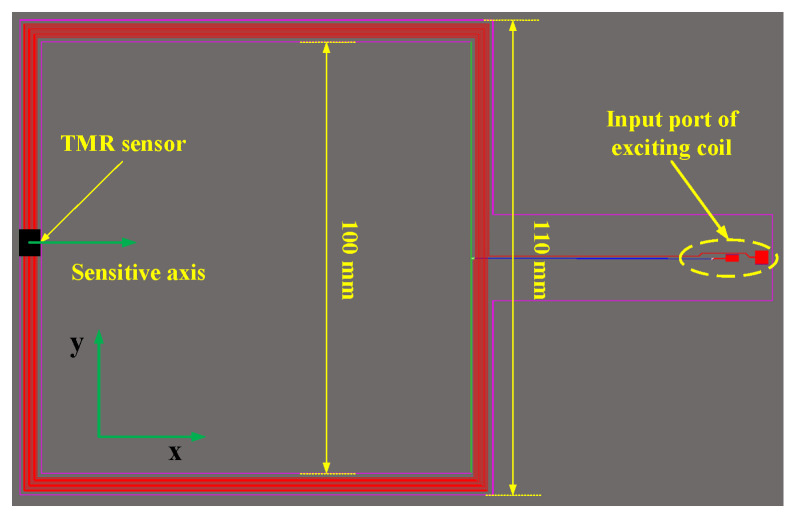
Dimension drawing of reference exciting coils.

**Figure 5 sensors-23-09507-f005:**
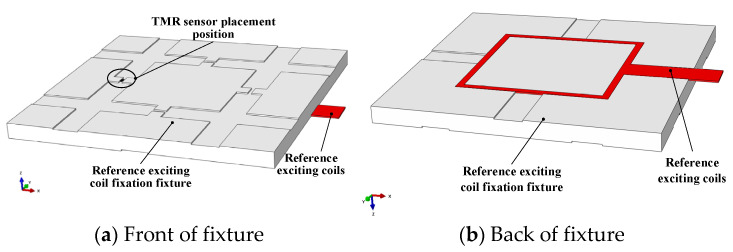
Installation diagram of TMR sensor above reference exciting coils.

**Figure 6 sensors-23-09507-f006:**
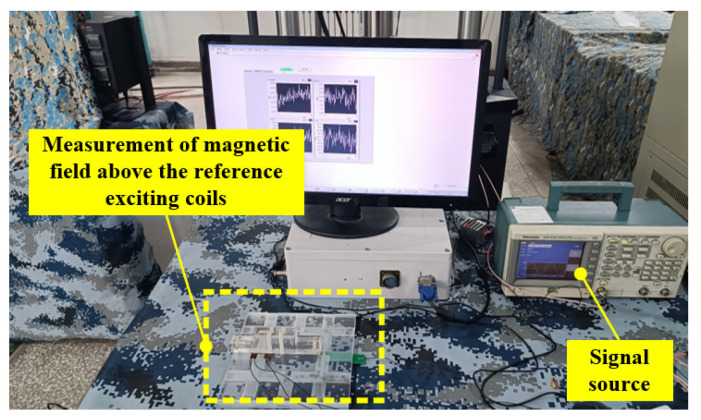
Magnetic field test in the area above the reference exciting coils.

**Figure 7 sensors-23-09507-f007:**
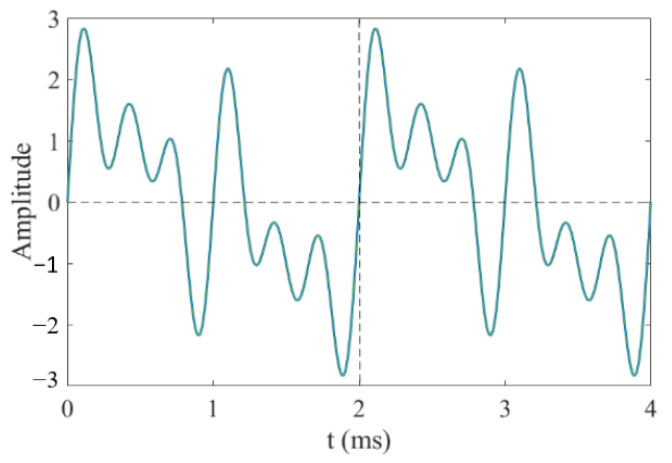
Synthetic waveform.

**Figure 8 sensors-23-09507-f008:**
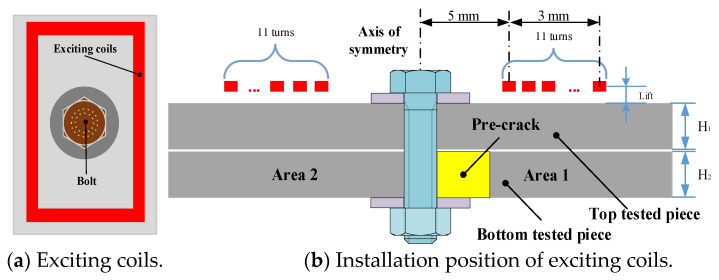
Installation form of exciting coils.

**Figure 9 sensors-23-09507-f009:**
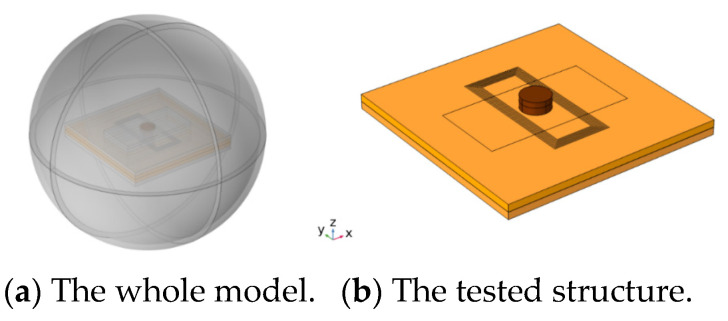
Three-dimensional simulation model.

**Figure 10 sensors-23-09507-f010:**
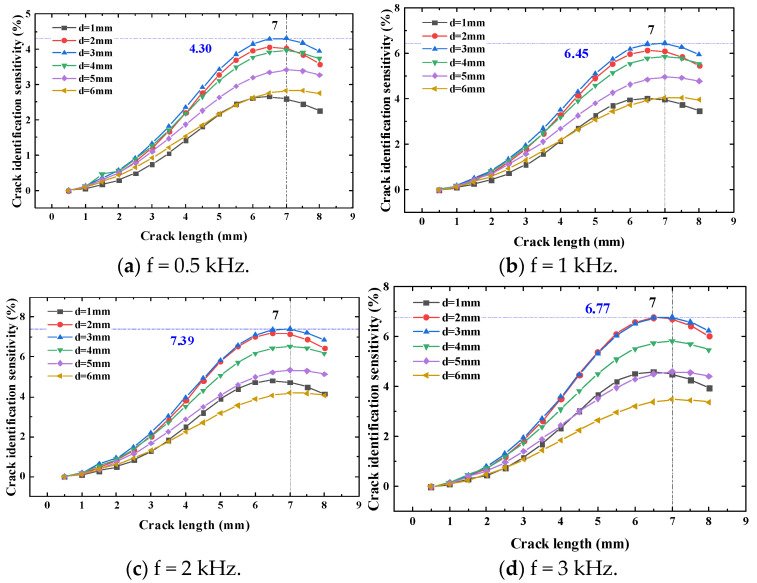
Crack identification sensitivity when the height of test points is 0.2 mm.

**Figure 11 sensors-23-09507-f011:**
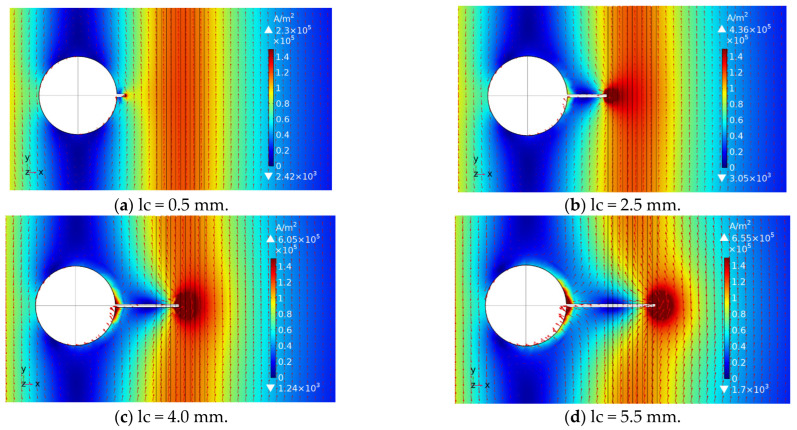
Disturbed eddy current caused by crack propagation.

**Figure 12 sensors-23-09507-f012:**
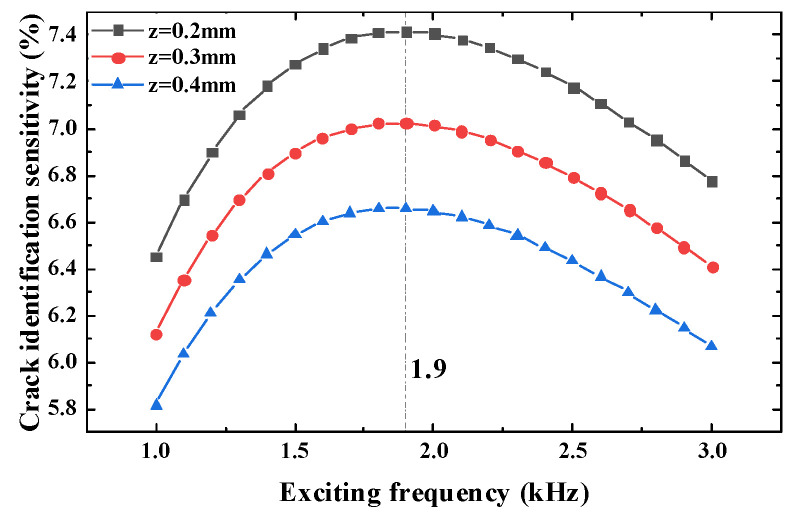
Optimal exciting frequency of simulation.

**Figure 13 sensors-23-09507-f013:**
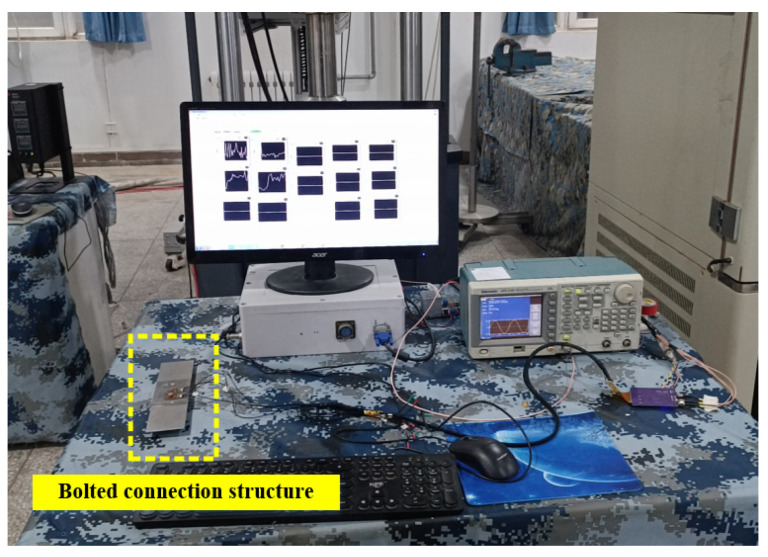
Experiment site of crack identification sensitivity.

**Figure 14 sensors-23-09507-f014:**
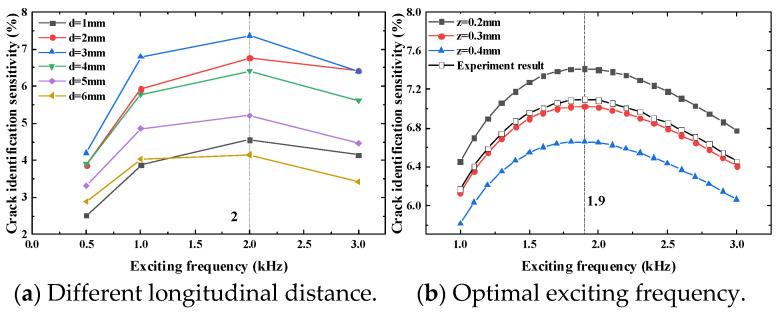
Crack monitoring test results, H_1_ = 2 mm.

**Figure 15 sensors-23-09507-f015:**
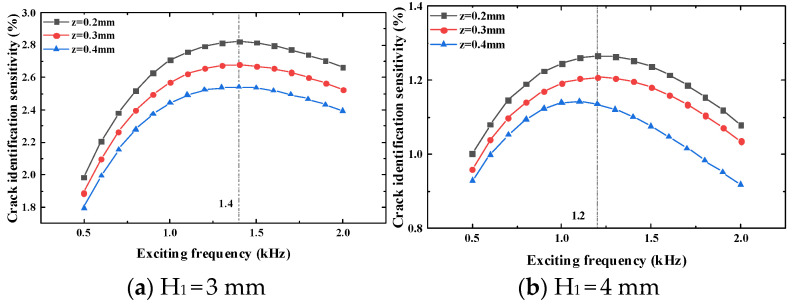
Simulation results of optimal the exciting frequency.

**Figure 16 sensors-23-09507-f016:**
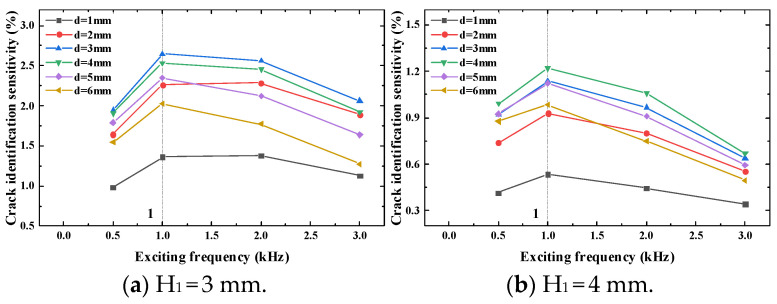
Crack identification sensitivity at different longitudinal distances.

**Figure 17 sensors-23-09507-f017:**
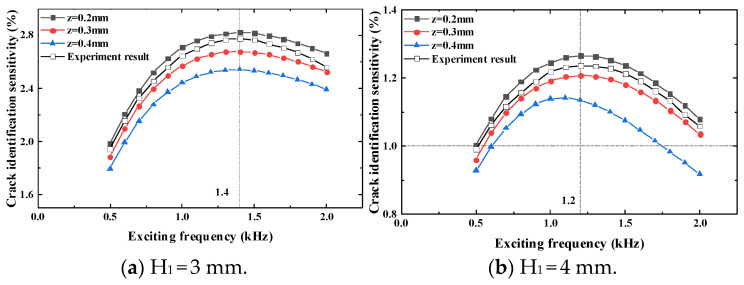
Experimental results of the optimal exciting frequency.

**Figure 18 sensors-23-09507-f018:**
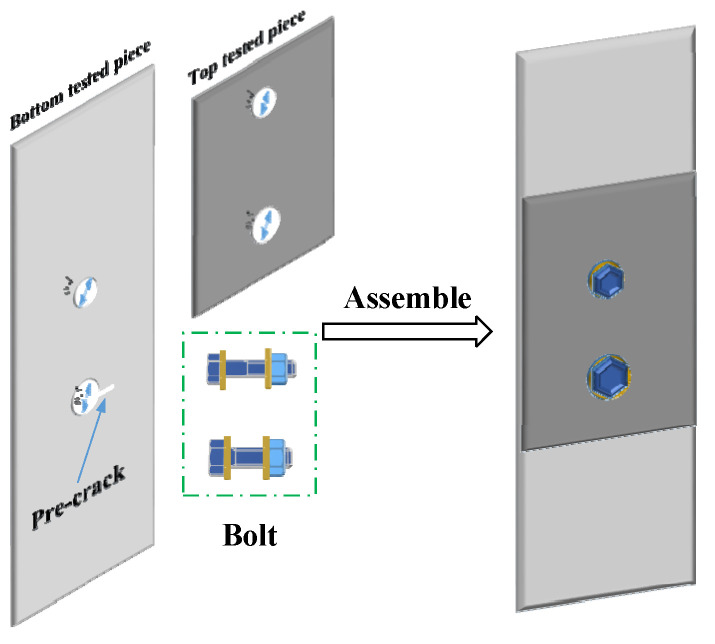
Metal connection structure.

**Figure 19 sensors-23-09507-f019:**
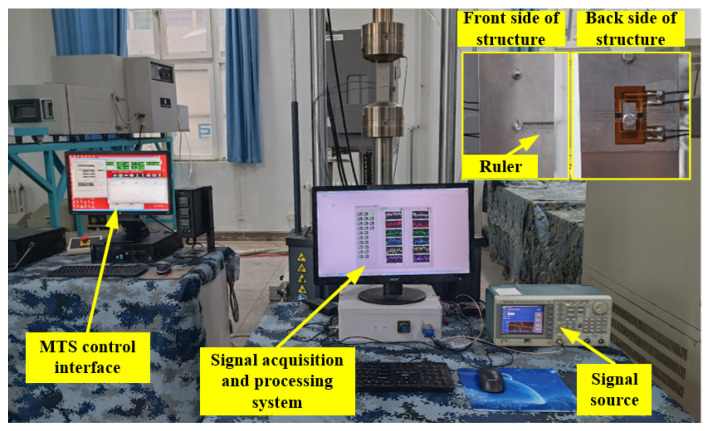
Internal fatigue crack monitoring test site.

**Figure 20 sensors-23-09507-f020:**
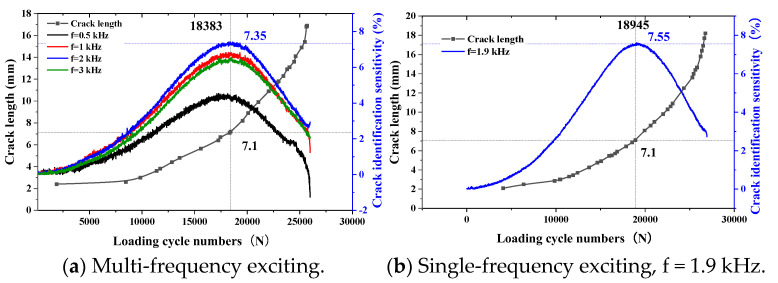
Monitoring results of internal crack (H_1_ = 2 mm).

**Figure 21 sensors-23-09507-f021:**
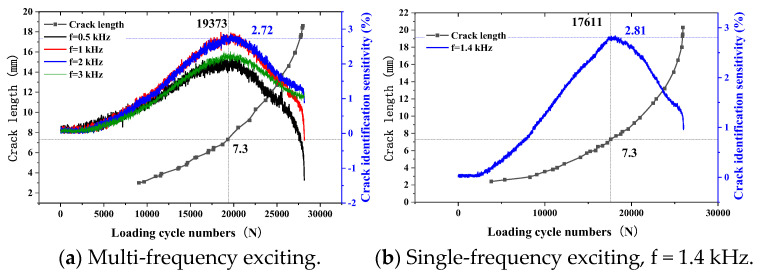
Monitoring results of internal crack (H_1_ = 3 mm).

**Figure 22 sensors-23-09507-f022:**
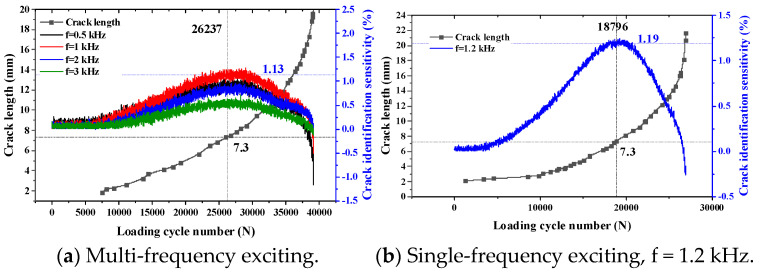
Monitoring results of internal crack (H_1_ = 4 mm).

**Table 1 sensors-23-09507-t001:** Simulation parameters.

Symbol	Physical Meaning	Value	Unit
rb	Bolt hole radius	2.5	mm
D	Distance between exciting coils	0.3	mm
l_f_	Lifting distance	0.1275	mm
H_1_	Thickness of top tested piece	2, 3, 4	mm
H_2_	Thickness of bottom tested piece	2	mm
σ	Conductivity of tested piece	1.74 × 10^7^	S/m
μr	Relative permeability of tested piece	1	1
f	Exciting frequency of the exciting current	0.5, 1, 2, 3	kHz
jw	Width of exciting coils	0.2	mm
jh	Thickness of exciting coil	0.0175	mm

**Table 2 sensors-23-09507-t002:** Crack identification sensitivity (z = 0.2 mm).

Exciting Frequency (kHz)	Maximum Crack Identification Sensitivity (%)	Longitudinal Distance (mm)
0.5	4.30	3
1	6.45	3
2	7.39	3
3	6.77	3

**Table 3 sensors-23-09507-t003:** Crack identification sensitivity (z = 0.3 mm).

Exciting Frequency (kHz)	Maximum Crack Identification Sensitivity (%)	Longitudinal Distance (mm)
0.5	4.08	3
1	6.11	3
2	7.00	3
3	6.40	3

**Table 4 sensors-23-09507-t004:** Crack identification sensitivity (z = 0.4 mm).

Exciting Frequency (kHz)	Maximum Crack Identification Sensitivity (%)	Longitudinal Distance (mm)
0.5	3.89	3
1	5.81	3
2	6.64	3
3	6.05	3

**Table 5 sensors-23-09507-t005:** Crack identification sensitivity (H_1_ = 3 mm, z = 0.2 mm).

Exciting Frequency (kHz)	Maximum Crack Identification Sensitivity (%)	Longitudinal Distance (mm)
0.5	1.98	3
1	2.71	3
2	2.66	3
3	2.14	3

**Table 6 sensors-23-09507-t006:** Crack identification sensitivity (H_1_ = 3 mm, z = 0.3 mm).

Exciting Frequency (kHz)	Maximum Crack Identification Sensitivity (%)	Longitudinal Distance (mm)
0.5	1.89	3
1	2.57	3
2	2.53	3
3	2.03	3

**Table 7 sensors-23-09507-t007:** Crack identification sensitivity (H_1_ = 3 mm, z = 0.4 mm).

Exciting Frequency (kHz)	Maximum Crack Identification Sensitivity (%)	Longitudinal Distance (mm)
0.5	1.80	3
1	2.44	3
2	2.39	3
3	1.92	3

**Table 8 sensors-23-09507-t008:** Crack identification sensitivity (H_1_ = 4 mm, z = 0.2 mm).

Exciting Frequency (kHz)	Maximum Crack Identification Sensitivity (%)	Longitudinal Distance (mm)
0.5	1.00	4
1	1.22	4
2	1.00	4
3	0.70	3

**Table 9 sensors-23-09507-t009:** Crack identification sensitivity (H_1_ = 4 mm, z = 0.3 mm).

Exciting Frequency (kHz)	Maximum Crack Identification Sensitivity (%)	Longitudinal Distance (mm)
0.5	0.96	4
1	1.17	4
2	0.95	4
3	0.66	3

**Table 10 sensors-23-09507-t010:** Crack identification sensitivity (H_1_ = 4 mm, z = 0.4 mm).

Exciting Frequency (kHz)	Maximum Crack Identification Sensitivity (%)	Longitudinal Distance (mm)
0.5	0.92	4
1	1.12	4
2	0.91	4
3	0.63	3

**Table 11 sensors-23-09507-t011:** Monitoring characteristics of cracks under top tested pieces with different thicknesses.

Thickness of Top Tested Piece (mm)	Multi-Frequency Exciting			Single-Frequency Exciting	
	Maximum Sensitivity (%)	Corresponding Crack Length (mm)	Corresponding Exciting Frequency (kHz)	Maximum Sensitivity (%)	Corresponding Crack Length (mm)
2	7.35	7.1	2	7.55	7.1
3	2.72	7.3	2.3	2.81	7.3
4	1.13	7.3	1	1.19	7.3

## Data Availability

Data are contained within the article.

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
