# Peer review of "A Flexible Eddy Current TMR Sensor for Monitoring Internal Fatigue Crack"

_sensors, 2023, doi:10.3390/s23239507_

Round 1

Reviewer 1 Report

Comments and Suggestions for Authors

TITLE: A flexible eddy current TMR sensor for monitoring internal fatigue crack

General remark:
This paper presents a sensor that uses a TMR mounted over a flexible layer. A lot of work has been done to demonstrate the performance of the device.

Punctual remarks:

1. Introduction, first line of the second paragraph: to say that conventional non-destructive testing technology is inefficient is a too strong affirmation. You must soften your affirmation by saying: “In some applications conventional non-destructive testing…”, or you may refer a specific application where the conventional technology is not efficient.

2. The standard depth of penetration in equation (1) is wrong.

3. Automatic references to other parts of the manuscript (ex.: figures) are not working.

4. Before figure 4 the authors refer the dimensions of a reference exciting coils. Those dimensions don’t agree with those in figure 4. 

5. On Table 1 the physical units of the represented data must be included.

6. In section 3.1 fem results for crack sensitivity are represented. The optimization of sensor sensitivity was obtained by simulation in different situations. Then these data were verified using a real experimental setup.

7. Finally one important question: The authors in the introduction section refer the structural health monitoring nature of the work presented. In that framework the bolt should not be removed for inspection. The authors must refer this issue in the revised paper.

Author Response

We are very grateful and excited to have been given the opportunity to revise our manuscript entitled “A flexible eddy current TMR sensor for monitoring internal fatigue crack” (No.: sensors-2669026). We carefully revised the manuscript considering the precious comments by reviewers and editor. Herein, we explain how we revised the paper based on those comments and recommendations. We want to extend our great appreciation of taking time and effect necessary to provide such insightful guidance.

The responses to the reviewers’ and editor’s comments are listed in the end. We tried our best to improve the manuscript and made some changes in the revised manuscript. The changes are marked in yellow in revised manuscript.

We appreciate for your and reviewers’ warm work earnestly, and hope that the correction will meet with approval. Once again, thank you so much for your help and looking forward to hearing from you soon.

Sincerely yours

Fei Yang

On behave of all authors.

Response to reviewer

 Comment 1:

Introduction, first line of the second paragraph: to say that conventional non-destructive testing technology is inefficient is a too strong affirmation. You must soften your affirmation by saying: “In some applications conventional non-destructive testing…”, or you may refer a specific application where the conventional technology is not efficient.

Response1:

Thanks for reviewer’s comment. I have revised the sentence according to your suggestion. And the changes are marked in yellow in the paper.

Comment 2:

The standard depth of penetration in equation (1) is wrong.

Response2:

Thanks for reviewer’s comment. I have revised the equation (1). And the changes are marked in yellow in the paper.

Comment 3:

Automatic references to other parts of the manuscript (ex.: figures) are not working.

Response3:

Thanks for reviewer’s comment. I have revised them according to your suggestion. And the changes are marked in yellow in the paper.

Comment 4:

Before figure 4 the authors refer the dimensions of a reference exciting coils. Those dimensions don’t agree with those in figure 4.

Response4:

Thanks for reviewer’s comment. I have changed it. The side length of the innermost square exciting coil is 102mm, and that of the outermost circle is 108 mm. The reference exciting coils are printed on the FR4 board with a thickness of 0.3mm, and the distance between the square exciting coils and the PCB edge is 1mm. And the changes are marked in yellow in the paper.

Comment 5:

On Table 1 the physical units of the represented data must be included.

Response5:

Thanks for reviewer’s comment. I have added them. And the changes are marked in yellow in the paper.

Comment 6:

In section 3.1 fem results for crack sensitivity are represented. The optimization of sensor sensitivity was obtained by simulation in different situations. Then these data were verified using a real experimental setup.

Response 6:

Thanks for reviewer’s comment. In order to further verify the correctness of the simulation, it is necessary to build a test system to carry out tests for verification.

Comment 7:

Finally one important question: The authors in the introduction section refer the structural health monitoring nature of the work presented. In that framework the bolt should not be removed for inspection. The authors must refer this issue in the revised paper.

Response 7:

Thanks for reviewer’s comment. The bolt can not be removed when the bolt hole edge crack is monitored. If the bolt is removed, on the one hand, it will introduce a large amount of work, on the other hand, it will also introduce new damage in the disassembly and installation process. The benefits of structural health monitoring will not be achieved. I have added it in the introduction, and the changes are marked in yellow in the paper.

Again, we appreciate very much the time and energy of Editor and Reviewers for the valuable suggestion and comments. Thank you for your consideration.

Sincerely

Fei Yang, and all co-authors

Reviewer 2 Report

Comments and Suggestions for Authors

The paper discusses the use of TMR sensors and exiting coils for detecting cracks in metals. This could be of interest and possibly be published.

The manuscript itself, however, does not meet the standard of Sensors due to the following reasons:

1.      The manuscript is by far too long. Many of the results (such as tables 2.6, sections 4.2.1 – 4.2.3, figures 10 b and c, 19, 21, 22, etc.) could be just presented as a part of a sentence.

2.      Almost all referrals are missing (Error! Reference source not found).

3.      One justification of the paper is given on page 5: “The magneto-resistive sensor selected in this paper is the TMR2901. The sensitivity of this sensor varies widely, which ranges from 20 mV/V/Oe to 27 mV/V/Oe. Therefore, it is difficult to ensure the consistency of the sensor in the application process.” A better way, however, would be probably to look for a better sensor such as the CT130 from Crocus.

4.      The circuit shown in Figure 1 has two TMR sensors. If I have understood correctly, there is only one sensor used. Wouldn’t it be much better to make differential measurements using both sensors?

5.      The most important part is missing: Is the proposed system better than the already existing techniques?

6.      In Figure 7, it cannot be seen that the period of the synthesized waveform is 2ms.

Comments on the Quality of English Language

Can be improved considerably

Author Response

We are very grateful and excited to have been given the opportunity to revise our manuscript entitled “A flexible eddy current TMR sensor for monitoring internal fatigue crack” (No.: sensors-2669026). We carefully revised the manuscript considering the precious comments by reviewers and editor. Herein, we explain how we revised the paper based on those comments and recommendations. We want to extend our great appreciation of taking time and effect necessary to provide such insightful guidance.

Combined with the comments put forward by the reviews, we have carefully revised and answered each article, hoping to make the manuscript better in the process of revision. We hope that the carefully revised manuscript based on the reviewer's comments can suitable for publication in Sensors.

The responses to the reviewers’ and editor’s comments are listed in the end. We tried our best to improve the manuscript and made some changes in the revised manuscript. The changes are marked in yellow in revised manuscript.

We appreciate for your and reviewers’ warm work earnestly, and hope that the correction will meet with approval. Once again, thank you so much for your help .Sincerely yours

Fei Yang

On behave of all authors.

Response to reviewer

Comments 1:

The manuscript is by far too long. Many of the results (such as tables 2.6, sections 4.2.1 – 4.2.3, figures 10 b and c, 19, 21, 22, etc.) could be just presented as a part of a sentence.

Response 1:

Thanks for reviewer’s comment. I think these charts can better represent the simulation and experimental results, so I hope to keep them.

Comments 2:

Almost all referrals are missing (Error! Reference source not found).

Response 2:

Thanks for reviewer’s comment. According to your suggestion, I have corrected them in paper, and the changes are marked in yellow.

Comments 3:

One justification of the paper is given on page 5: “The magneto-resistive sensor selected in this paper is the TMR2901. The sensitivity of this sensor varies widely, which ranges from 20 mV/V/Oe to 27 mV/V/Oe. Therefore, it is difficult to ensure the consistency of the sensor in the application process.” A better way, however, would be probably to look for a better sensor such as the CT130 from Crocus.

Response 3:

Thanks for reviewer’s comment. The inconsistency of the sensitivity of the magneto-resistive sensor has a great influence on the test results. In order to improve the accuracy of the test results, a magneto-resistive sensor with good sensitivity consistency can be selected, but at the same time, the size, price, working conditions and other factors of the magneto-resistive sensor should be considered. In this paper, in order to avoid the error introduced by the inconsistency of the magneto-resistive sensor, a method using the reference excitation coil is used to correct it, which has the advantages of low cost and high accuracy.

Comments 4:

The circuit shown in Figure 1 has two TMR sensors. If I have understood correctly, there is only one sensor used. Wouldn’t it be much better to make differential measurements using both sensors?

Response 4:

Thanks for reviewer’s comment. The FEC-TMR sensor consists of rectangular exciting coils and two TMR sensors. These two sensors are symmetrically distributed, and in order to avoid duplication of work when conducting the experimental study, only one side of the TMR sensor is used to carry out the study. I will make a supplementary explanation of this part in the article, and the changes are marked in yellow.

Comment 5:

The most important part is missing: Is the proposed system better than the already existing techniques?

Response 5: 

Thanks for reviewer’s comment. As I described in the introduction section, there are many researches on bolt hole edge crack monitoring at the present stage, but most of them focus on surface cracks, and most of them do not consider the influence of bolt extrusion and the monitoring of internal cracks in multi-layer metal connection structures. In this paper, by reducing the excitation frequency and using the TMR sensor as the induction unit, the internal crack monitoring of the multi-layer metal connection structure is realized.

Comments 6:

In Figure 7, it cannot be seen that the period of the synthesized waveform is 2ms.

Response 6:

Thanks for reviewer’s comment. I have change it. It cannot be seen that the period of the synthesized waveform is 2 ms, and the changes are marked in yellow.

Again, we appreciate very much the time and energy of Editor and Reviewers for the valuable suggestion and comments. Thank you for your consideration.

Sincerely

Fei Yang, and all co-authors

Reviewer 3 Report

Comments and Suggestions for Authors

The paper presents a flexible eddy current TMR (FEC-TMR) sensor for monitoring internal cracks in metal joint structures. The study is commendable in its approach to combine simulation and experimental validation to assess the sensor's performance. However, there are several areas where the paper could be improved to enhance its clarity and scientific rigor.

The introduction should provide more context on the importance of monitoring internal cracks in metal joint structures and how it relates to current challenges in industry or safety.

Additionally, it should clearly state the research objectives and hypotheses, which are somewhat implicit in the current version.

The paper mentions the establishment of a finite element model for sensor analysis but lacks detail on the specifics of this model. What software or tools were used for the simulation, and what parameters were considered?

It would be beneficial to provide a clear step-by-step description of how the sensor's crack identification sensitivity was analyzed and how the optimal location and exciting frequency were determined.

The paper discusses testing the optimal longitudinal spacing and exciting frequency, but the methods, equipment, and specific results of these experiments are not detailed. This information is essential to replicate the study.

The results of the experiments are said to be consistent with the simulation, but quantitative data and statistical analysis should be provided to demonstrate this consistency.

The paper mentions that "the experimental results are consistent with the simulation results," which is a critical finding. However, more discussion is needed on why this consistency is significant and how it impacts the sensor's practical utility.

When discussing the crack identification sensitivity, it is crucial to provide quantitative data, graphs, or tables to illustrate the decrease in sensitivity with increasing depth.

The conclusion should provide a concise summary of the key findings and their implications.

The paper could benefit from suggesting potential applications or future research directions based on the results presented.

The paper lacks figures or tables to visually represent the simulation and experimental data. Including these visuals can greatly enhance the clarity of the presentation.

The paper is generally well-written, but there are minor grammar and language issues throughout. Proofreading and editing for clarity are recommended.

The structure of the paper could be improved for better flow and logical progression. Consider revising the organization to make it more reader-friendly.

Overall, the concept and methodology of the FEC-TMR sensor are intriguing, but the paper needs to provide more details on the methods and results to make it a valuable contribution to the field. Additionally, enhancing clarity and providing quantitative data will strengthen the paper's impact.

Author Response

Dear  Reviewers:

We are very grateful and excited to have been given the opportunity to revise our manuscript entitled “A flexible eddy current TMR sensor for monitoring internal fatigue crack” (No.: sensors-2669026). We carefully revised the manuscript considering the precious comments by reviewers and editor. Herein, we explain how we revised the paper based on those comments and recommendations. We want to extend our great appreciation of taking time and effect necessary to provide such insightful guidance.

The responses to the reviewers’ and editor’s comments are listed in the end. We tried our best to improve the manuscript and made some changes in the revised manuscript. The changes are marked in yellow in revised manuscript.

We appreciate for your and reviewers’ warm work earnestly, and hope that the correction will meet with approval. Once again, thank you so much for your help and looking forward to hearing from you.

Sincerely yours

Yuting He

On behave of all authors.

Response to reviewer

Comments 1:

The introduction should provide more context on the importance of monitoring internal cracks in metal joint structures and how it relates to current challenges in industry or safety.

Response 1:

Thanks for reviewer’s comment. I have added more context on the importance of monitoring internal cracks in metal joint structures. And the changes are marked in yellow in introduction.

Comments 2:

Additionally, it should clearly state the research objectives and hypotheses, which are somewhat implicit in the current version.

Response 2:

Thanks for reviewer’s comment. I have clearly stated the research objectives and hypotheses in the last paragraph of the introduction. And the changes are marked in yellow in introduction.

Comments 3:

The paper mentions the establishment of a finite element model for sensor analysis but lacks detail on the specifics of this model. What software or tools were used for the simulation, and what parameters were considered?

Response 3:

Thanks for reviewer’s comment. The finite element simulation model was established using COMSOL multi-physics finite element software. And the simulation parameters are shown in table 1.

Comments 4:

It would be beneficial to provide a clear step-by-step description of how the sensor's crack identification sensitivity was analyzed and how the optimal location and exciting frequency were determined.

Response 4:

Thanks for reviewer’s comment. According to your suggestions, the sensor's crack identification sensitivity is descripted in section 2.3. And the optimal location and exciting frequency are determined by equation (8) according to simulation results in section 3.

Comments 5:

The paper discusses testing the optimal longitudinal spacing and exciting frequency, but the methods, equipment, and specific results of these experiments are not detailed. This information is essential to replicate the study.

Response 5:

Thanks for reviewer’s comment. The methods and equipment are descripted in section 2.4. And the specific results of these experiments are descripted in figure 14.

Comments 6:

The results of the experiments are said to be consistent with the simulation, but quantitative data and statistical analysis should be provided to demonstrate this consistency.

Response 6:

Thanks for reviewer’s comment. For the experiment results, it can be attained that when the crack propagates to about 7mm, the crack identification sensitivity of the TMR sensor reaches the maximum value, which is consistent with the simulation results. And the preliminary quantification of the crack length can be judged according to the maximum sensitivity of the TMR sensor. I added it in the paper, and the changes are marked in yellow in the paper.

Comments 7:

The paper mentions that "the experimental results are consistent with the simulation results," which is a critical finding. However, more discussion is needed on why this consistency is significant and how it impacts the sensor's practical utility.

Response 7:

Thanks for reviewer’s comment. It can be attained that the preliminary quantification of the crack length can be judged according to the maximum sensitivity of the TMR sensor.

Comments 8:

When discussing the crack identification sensitivity, it is crucial to provide quantitative data, graphs, or tables to illustrate the decrease in sensitivity with increasing depth.

Response 8:

Thanks for reviewer’s comment. The simulation results and experiment results are shown respectively in Table 2-Table 10 and Table 11.

Comments 9:

The conclusion should provide a concise summary of the key findings and their implications.

Response 9:

Thanks for reviewer’s comment. According to your suggestion, the conclusion is revised. And the changes are marked in yellow in the paper.

Comments 10:

The paper could benefit from suggesting potential applications or future research directions based on the results presented.

Response 10:

Thanks for reviewer’s comment. The results of this paper can be used to monitor the bolted connection structures that are difficult to detect and ensure the safe operation of the structures in engineering applications. And I have added it in the conclusion.

Comments 11:

The paper lacks figures or tables to visually represent the simulation and experimental data. Including these visuals can greatly enhance the clarity of the presentation.

Response 11:

Thanks for reviewer’s comment. The simulation and experimental data are shown in table 2-table 11.

Comments 12:

The paper is generally well-written, but there are minor grammar and language issues throughout. Proofreading and editing for clarity are recommended.

Response 12:

Thanks for reviewer’s comment. According to your suggestions, I have refined the article.

Comments 13:

The structure of the paper could be improved for better flow and logical progression. Consider revising the organization to make it more reader-friendly.

Response 13:

Thanks for reviewer’s comment. According to your suggestion, I have changed the structure of the paper.

Comments 14:

Overall, the concept and methodology of the FEC-TMR sensor are intriguing, but the paper needs to provide more details on the methods and results to make it a valuable contribution to the field. Additionally, enhancing clarity and providing quantitative data will strengthen the paper's impact.

Response 14:

Thanks for reviewer’s comment. Thank you for your recognition. I have modified the sentence and adjusted the structure of the paper.

Again, we appreciate very much the time and energy of Editor and Reviewers for the valuable suggestion and comments. Thank you for your consideration.

Sincerely

Fei Yang, and all co-authors
